# Clinical epidemiology and high genetic diversity amongst *Cryptococcus* spp. isolates infecting people living with HIV in Kinshasa, Democratic Republic of Congo

**Bive Zono Bive**[1,2]*, **Rosalie Sacheli**[2,3°], **Hippolyte Situakibanza Nani-Tuma**[4‡], **Pius Kabututu Zakayi**[1‡], **Alex Ka**[5‡], **Marcel Mbula Mambimbi**[4‡], **Gaultier Muendele**[6‡], **Raphael Boreux**[2‡], **Nicole Landu**[7‡], **Celestin Nzanzu Mudogo**[1‡], **Pierre-Robert M'Buze**[8‡], **Michel Moutschen**[9‡], **Wieland Meyer**[5,10‡], **Georges Mvumbi Lelo**[1°], **Marie-Pierre Hayette**[2,3°]

1 Faculty of Medicine, Department of Basic Sciences, Molecular Biology Service, University of Kinshasa, Kinshasa, The Democratic Republic of Congo, 2 Centre for Interdisciplinary Research on Medicines, University of Liege, Liege, Belgium, 3 National Reference Centre for Mycosis, University Hospital Centre of Liege, Liege, Belgium, 4 Faculty of Medicine, Department of Internal Medicine/Department of Tropical Medicine, Infectious Diseases Service, University of Kinshasa, Kinshasa, The Democratic Republic of Congo, 5 Molecular Mycology Research Laboratory, Faculty of Medicine and Health, Centre for Infectious Diseases and Microbiology, Sydney Medical School, Westmead Clinical School, Marie Bashir Institute for Infectious Diseases and Biosecurity, University of Sydney, Westmead Hospital-Research and Education Network, Westmead Institute for Medical Research, Sydney, Australia, 6 Internal Medicine Department, Advanced HIV Disease Management Unit, Centre Hospitalier Mère et Enfant de NGABA, Kinshasa, The Democratic Republic of Congo, 7 Internal Medicine Department, Advanced HIV Disease Management Unit, Centre Médical et Evangélique Révérend LUYINDU, Kinshasa, The Democratic Republic of Congo, 8 Internal Medicine Department, Advanced HIV Disease Management Unit, Centre Hospitalier Roi Baudouin 1er, Kinshasa, The Democratic Republic of Congo, 9 Department of Infectious Diseases and General Internal Medicine, University Hospital Centre of Liege, Liege, Belgium, 10 Curtin Medical School, Curtin University, Perth, Australia

° These authors contributed equally to this work.
‡ These authors also contributed equally to this work.
* bive.zono@unikin.ac.cd

**Data Availability Statement:** Clinical and genomic data from Cryptococcus neoformans infections

## Abstract

Neuromeningeal cryptococcosis (NMC) is a life-threatening opportunistic infection in advanced HIV disease patients (AHDP). It is caused by *Cryptococcus* spp. complexes and mainly occurs in sub-Saharan Africa. In this study, we performed molecular characterization and antifungal susceptibility profiling of *Cryptococcus* isolates from AHDP in Kinshasa (DRC). Additionally, we investigated a possible association between NMC severity factors and the *Cryptococcus neoformans* (*Cn*) multilocus sequence typing (MLST) profiles. We characterized the isolates using PCR serotyping, MALDI-TOF MS, internal transcribed spacer (ITS) sequencing, and MLST. Susceptibility testing for the major antifungal drugs was performed according to the EUCAST guidelines. Parameters associated with NMC severity, such as hypoglycorrhachia (< 50 mg/dL), increased cerebral spinal fluid opening pressure (> 30 cm $H_2O$), and poor therapeutic outcome were compared with the *Cn* MLST sequences type (ST). Twenty-three out of 29 *Cryptococcus* isolates were identified as serotype A using PCR serotyping (79.3%; 95% IC: 65.5–93.1), while six (20.7%; 95% IC: 6.9–

were submitted and integrated into the International Multilocus Sequence Typing (MLST) Data base (https://mlst.mycologylab.org/). The novel Sequence Type (ST) isolated for the first time in our study (not yet reported from other parts of the world) was assigned as ST659 and can be found as such in the database. We attach the submission sheet for easy access. Concerning Cutaneotrichospora curvatus/Papiliotrema laurentii infection data, the genomic (and clinical) information of the ITS-2 sequences as submitted to GenBank have also been uploaded as supporting information file. In addition, all ITS-2 sequences extracted in the study have been provided as supplementary information.

**Funding:** This study was funded by ARES-Belgium (Académie de Recherche et d'Enseignement Supérieur), through an institutional support programme to the University of Kinshasa (under the grant number: COOP-CONV-18-004, AI ARES UNIKIN). The study's funder had no role in study design, data collection, data analysis, data interpretation, or writing of the report.

**Competing interests:** The authors have declared that no competing interests exist.

34.5) were not serotypable. The 29 isolates were identified by ITS sequencing as follows: *Cryptococcus neoformans* (23/29, 79.3%), *Cutaneotrichosporon curvatus* (previously called *Cryptococcus curvatus*) (5/29, 17.2%), and *Papiliotrema laurentii* (*Cryptococcus laurentii*) (1/29, 3.5%). Using the ISHAM MLST scheme, all *Cn* isolates were identified as molecular type VNI. These comprised seven different STs: ST93 (n = 15), ST5 (n = 2), ST53 (n = 1), ST31 (n = 1), ST4 (n = 1), ST69 (n = 1), and one novel ST that has not yet been reported from other parts of the world and was subsequently assigned as ST659 (n = 2). Of the included strains, only *Papiliotrema laurentii* was resistant to amphoterin B (1/29, 3.5%), 6.8% (2/29) were resistant to 5-flucytosine (the single *Papiliotrema laurentii* strain and one *Cryptococcus neoformans* isolate), and 13.8% (4/29) to fluconazole, including two of five (40%) *Cutaneotrichosporon curvatus* and two of 23 (8.7%) *C. neoformans* strains. We found a significative association between poor therapeutic outcome and a non-ST93 sequence type of causative strains (these concerned the less common sequence types: ST53, ST31, ST5, ST4, ST659, and ST69) (87.5% versus 40%, $p$ = 0.02). Molecular analysis of *Cryptococcus* spp. isolates showed a wide species diversity and genetic heterogenicity of *Cn* within the VNI molecular type. Furthermore, it is worrying that among included strains we found resistances to several of the commonly used antifungals.

## Introduction

Among opportunistic infections encountered during HIV/AIDS, neuromeningeal cryptococcosis (NMC) causes 15% of deaths. Seventy-five per cent of deaths occur in sub-Saharan Africa, where the Democratic Republic of Congo (DRC) is also located. In this region, the annual mortality from this invasive fungal infection is estimated at 135,900 per year, making it one of the leading causes of death from infectious diseases [1].

The DRC is the second-largest African country with an area of 2,345,409 km$^2$ and a population of approximately 95,326,000. More than 60% of the population lives in rural areas. Its borders of about 9,165 kilometres are shared with nine neighbouring countries, namely: the Republic of Congo, Central African Republic, South Sudan, Uganda, Rwanda, Burundi, Tanzania, Zambia, and Angola [2].

The 2021 report of the United Nations Programme on AIDS (UNAIDS) in DRC estimated that 510,000 people were living with HIV (PLHIV) in 2020. Of these, approximately 25% (130,000) were not on antiretroviral treatment (ART) and therefore at risk of all the recognised complications of advanced HIV disease (AHD), such as cryptococcosis. In this HIV population at high risk of cryptococcosis, we include PLHIV in treatment failure and those on ART but lost to follow-up. Thus, in Kinshasa, the prevalence of NMC has previously been estimated at 8.8%, with a mortality rate of one in three patients [3,4].

The *Cryptococcus neoformans/C. gattii* species complexes (*Cn/Cg*) are the main etiological agents of cryptococcosis. Based on epidemiological, pathobiology, geographical distribution, ecological niches, clinical presentation, therapeutic, and genetic differences [5], seven distinct haploid species and four hybrid species are distinguished [6]. Apart from the *Cn/Cg* species complexes, others species previously described as non-*neoformans/gattii Cryptococcus* are also (rarely) associated with cryptococcal infection; these are now classified as members of other genera in the *Tremellomycetes* subphylum. This is the case for *Papiliotrema laurentii* (previously named *Cryptococcus laurentii*), *Cutaneotrichosporon curvatus* (*Cryptococcus curvatus*), *Naganishia albidus* (*Cryptococcus albidus*), *Naganishia diffluens* (*Cryptococcus diffluens*) [7–9].

Infections caused by specific *Cryptococcus* spp., such as *C. gattii* sensu lato (s.l), require more intensive therapy than those caused by *C. neoformans* s.l. Furthermore, several of the non-*Cryptococcus* species that cause cryptococcosis (according to the old taxonomy) exhibit intrinsic resistance to some of the commonly used antifungal agents [10–12]. Hence, surveillance data on circulating *Cryptococcus* spp. molecular types and their antifungal susceptibility profiles could inform the establishment of local therapeutic guidelines.

As the association of *Cn/Cg* molecular type with the antifungal susceptibility profile has previously been established, it is also opportune to verify the association of molecular types, or even MLST sequence types (ST), with clinical presentation. Therefore, we investigated whether NMC severity factors could be linked with the sequence type of the causative strain [13].

Here we describe the NMC clinical epidemiology amongst advanced HIV disease patients (AHDP), the molecular characterization of strains, as well as the antifungal susceptibility of *Cryptococcus* spp. isolates. In addition, the association between NMC severity factors and *Cryptococcus neoformans* MLST sequence types was statistically analysed.

The findings of this study will allow programme decision-makers and health care providers to consider the local specificities of this deadly mycosis, in terms of epidemiological magnitude, circulating species, clinical severity and antifungal susceptibility. In addition, they will serve as a reference in molecular epidemiology and antifungal susceptibility for future surveillance studies, which should be performed regularly.

## Materials and methods

### Study design, patients and samples

A cross-sectional study was conducted in three Kinshasa public hospitals supported by Doctors without Borders-Belgium (MSF), from 1 February 2019 to 29 February 2020. Thus, 278 patients were included and among them, NMC was diagnosed based on the cryptococcal antigens (CrAg) detection and/or the presence of yeasts cells detected by India ink staining and/or by culture.

### Biological analysis

The CrAg detection was carried out in the cerebral spinal fluid (CSF) of each included patient, using the CrAg LFA IMMY test (Immuno-mycologic, Norman, OK, USA). Direct staining with India ink in the CSF was also carried out, and the CSF was cultured on Sabouraud Dextrose Agar-Chloramphenicol medium (SDA-C, bioMérieux, France) at 30˚C for 48 to 72h. The qualitative Pandy test was performed for detecting elevated protein levels in CSF, as previously described [14].

**Identification by MALDI-TOF MS.**   Fungal identification was performed using MALDI-TOF MS on a Bruker Microflex instrument (Bruker Daltonics GmbH, Bremen, Germany). From the culture on SDA-C, a partial extraction procedure was performed by adding 1 μL of 70% formic acid to the sample on a target plate (MSP 96 BC ground steel target; Bruker Daltonics). Then, 1 μL of saturated cyano-4-hydroxycinnamic acid solution (HCCA matrix; Bruker Daltonics) was added. Each microorganism tested was spotted twice on the same target plate. Measurements were performed with the Flex control V3.4 software (Bruker Daltonics), using default settings and BD 8326 (version V. 9.0) as a reference library [15]. The following criterion was used for reliable identification of fungal species: an MS Score ≥ 1.5 and the 3 first results identical and consistent with the appearance of the colonies on agar (with some adjustments according to the scores profile for difficult cases)

**Molecular analysis.**   *DNA extraction*. Genomic DNA was extracted from the fresh 24-hour cultures using the NucleoSpin blood quick pure kit (Macherey-Nagel, Düren,

Germany). Two preliminary steps were added to the manufacturer's protocol, namely bead-beating and thermal shock. In a 2mL tube containing 0.5mm glass beads (Roche Diagnostics GmbH, Penzberg, Germany), colonies were mixed with 350 μL lysis buffer (Promega Corporation, Madison, Wisconsin, USA). The mixture was vortexed five times at 6000 revolutions per minute for 40 seconds (bead-beating). Between each pass, the tube was cooled between -20°C and 1°C for 30 seconds in a Nalgene microtube cooler container (Dutscher, Bernolsheim, France) for a thermal shock.

*Serotyping PCR.* A classical serotyping PCR designed for *Cn/Cg* species complexes was performed according to the protocol described by Ito-Kuwa *et al.* [16].

*ITS sequencing.* The ITS2 region of the rRNA gene cluster was amplified using the ITS86 forward primer 5′GTGAATCATCGAATCTTTGAA 3′ and ITS4 reverse primer 5′TCCTCC GCTTATTGATATGC 3′ [17]. The amplified products were purified using the clean Seq Agencourt kit (Beckman Coulter Life Sciences, Indianapolis, Indiana, USA). Sequencing was done on an ABI 3500/3500XL automate (Applied Biosystems, Thermo Fisher Scientific, Waltham, Massachusetts, USA). Bidirectional sequence data were generated after purification using the BigDye terminator sequencing kit (Applied Biosystems). Sequences generated by the software ABI Sequence Scanner V.1.0 (Applied Biosystems) were then compared to the CBS database by using The BioloMICS database software (https://wi.knaw.nl/page/Pairwise_alignment), which comprises several databases including Genbank. Only results that repeated the same identification at least three times and had a similarity score greater than 95% were considered valid.

*Multilocus sequence typing.* Multilocus sequence typing (MLST) was performed using the International Society of Human and Animal Mycology (ISHAM) consensus scheme for the *Cn/Cg* species complexes; including six unlinked housekeeping loci (GPD1, LAC1, URA5, SOD1, CAP59, and PLB1) and the non-coding region IGS1 [18]. After DNA extraction, samples were sequenced using Illumina HiSeq (Illumina, San Diego, California, USA) as previously described [19], and the raw contigs sequences were paired, removed of duplicate reads, and trimmed using Geneious Prime 64_2021_1 (https://www.geneious.com). Then, the MLST loci were extracted by mapping to the reference sequences of each MLST locus from the online ISHAM MLST fungal database (https://mlst.mycologylab.org/) and each allele type (AT) was assigned using the same online database. The ATs combination defines the sequences type (ST) which in most cases corresponds to the molecular type of isolate.

Minimum expansion trees were generated using the geoBURST algorithm to determine the STs clonal complexes (CCs) representing linked and closely related clusters (https://online. phyloviz.net). Apart from STs of *C. neoformans* s.l and *C. gattii* s.l reference strains (WM) included in the analysis, we also inserted MLST sequences of the single *C. neoformans* strain (ZS) previously isolated from a Congolese infected patient and recorded in the International Fungal MLST Database (DRC, previously called Zaire).

**Antifungal susceptibility testing.** Antifungal drugs' minimal inhibitory concentrations (MICs) were determined according to EUCAST guidelines (European Committee on Antimicrobial Susceptibility Testing) E.Def 7.3.1 [20]. Suspensions of 0.5 McFarland standard were prepared and diluted 1:10 with sterile distilled water (Sensititre demineralized water, Thermo Fisher Scientific). Plates contained the following concentration ranges: 0.008–8 mg/L for amphotericin B, and 0.06–64 mg/L for both 5-flucytosine and fluconazole. The reading of the MIC50 value (drug concentration resulting in 50% inhibition of microorganisms) for 5-flucytosine and fluconazole, and MIC90 for amphotericin B, was done according to EUCAST recommendations, both visually and using an automated reading at 405 nm with a Multiscan FC spectrophotometer (Thermo Fisher Scientific). *Candida parapsilosis* ATCC 22019 and *Candida krusei* ATCC 6258 were used as the quality control strains for the tests. The interpretation

criteria for amphotericin B were those defined in the antifungals EUCAST breakpoint tables' version 10.0: susceptible, $\leq 1$ mg/L; resistant, $> 1$ mg/L. Since no EUCAST breakpoints were defined for fluconazole and 5-flucytosine, interpretation criteria were based on the epidemiological cut-off values for *in vitro* susceptibility testing provided by the Clinical and Laboratory Standards Institute (CLSI) for *C. neoformans* (AFLP1/VNI) as follows: for both fluconazole and 5-flucytosine, sensitive, $\leq 8$ mg/L; resistant, $> 8$ mg/L [21].

## Statistical analysis

Statistical analyses were carried out using R-cmdr version 2.6–1 (R Foundation for Statistical Computing, Vienna, Austria). Missing data were considered completely random and the available data were analysed. The continuous variables were summarised as mean ± standard deviation and compared using Student's t-test. The proportions and their respective 95% confidence intervals were calculated for the categorical data. The main outcome variable was the NMC diagnosis. This variable was compared to other variables of the same category using Pearson's chi-square test or Fisher's exact test if the expected values were less than five. Raised CSF opening pressure ($> 30$ cm $H_2O$), hypoglycorrhachia ($< 50$ mg/dL) and poor therapeutic outcome were considered in performing the association analysis between the NMC severity factors and the identified ST-MLST profile. Thus, two isolates categories were formed according to the STs-MLST profile: the main ST isolated (ST93) on the one hand, and the other STs on the other hand. All tests were two-tailed and a $p < 0.05$ was considered statistically significant.

Comparative data between PLHIV with *Cryptococcus neoformans* versus *Cutaneotrichosporon curvatus* and *Papiliotrema laurentii* meningitis are presented in another published paper.

## Ethical considerations

This work was carried out in strict compliance with ethical rules, with the approval of the Ethics Committee of the Public Health School of the Faculty of Medicine of the University of Kinshasa under the approval number ESP/CE/071/2019. All patients included in this study were informed of the risks associated with the study and gave their written informed consent to participate. Anonymity was guaranteed and the data collected was kept and handled by the research team alone.

## Results

Among 278 advanced HIV disease patients (AHDP) who were hospitalized with suspected cryptococcosis and underwent lumbar puncture during the study period, 66 (23.7%, 95% CI: 18.7–28.8) had NMC. The NMC prevalence was similar in males (24.8%, 95% CI: 14.9–34.7, 25/101 included males) and in females (23.2%, 95% CI: 15.7–30.7, 41/177 included females).

## Patients' characteristics

The demographic and clinical characteristics of the patients are presented in Table 1. The mean age of the included patients was 42.2±12.1 years old. Most of them concerned females (63.7%), married or cohabitating (49.3%), and had a secondary education level (55.8%). Regarding the HIV clinical stage of patients before NMC diagnosis, NMC tended to develop during stage IV (95.5%, $p = 0.0008$). AHDP with NMC had four times higher probability to present headaches (OR 3.8; IC 95%: 1.7–8.8; $p = 0.0001$), three times higher probability to

**Table 1. Demographic and clinical characteristics of patients.**

| Characteristics[1] | Overall data (%)[2] | NMC | | p-value | Crude OR (95% CI) |
|---|---|---|---|---|---|
| | | No (%) | Yes (%) | | |
| Demographic characteristics | | | | | |
| Mean age ± SD (years) (n = 278) | 42.2 ± 12.1 | 42.8 ± 12.0 | 40.2 ± 12.4 | 0.1 | |
| Female sex (n = 278) | 177 (63.67) | 136 (64.2) | 42 (62.1) | 0.7 | |
| Marital status (n = 278) | | | | 0.6 | |
| Single | 89 (32.01) | 71 (33.5) | 18 (27.3) | | |
| Married/cohabitating | 137 (49.28) | 102 (48.1) | 35 (53.0) | | |
| Divorced/widower | 52 (18.7) | 39 (18.4) | 13 (19.7) | | |
| Education level attained (n = 278) | | | | 0.6 | |
| None/primary | 69 (24.82) | 52 (24.5) | 17 (25.8) | | |
| Secondary | 155 (55.76) | 121 (57.1) | 34 (51.5) | | |
| Higher education/university | 54 (19.42) | 39 (18.4) | 15 (22.7) | | |
| Clinical characteristics | | | | | |
| HIV clinical stage (n = 242) | | | | 0.0008 | |
| Stage I | 1 (0.41) | 1 (0.5) | 0 (0.0) | | |
| Stage II | 2 (0.83) | 2 (0.9) | 0 (0.0) | | |
| Stage III | 51 (21.07) | 48 (22.6) | 3 (4.5) | | |
| Stage IV | 224 (92.6) | 161 (75.9) | 63 (95.5) | | |
| Headaches (n = 269) | 174 (64.68) | 120 (58.8) | 54 (84.4) | 0.0001 | 3.8 (1.7–8.8) |
| Fever (˚C) (n = 269) | 179 (66.54) | 141 (68.8) | 38 (59.4) | 0.1 | |
| Weight loss (n = 269) | 144 (53.53) | 107 (52.2) | 37 (57.8) | 0.4 | |
| Consciousness disorder (n = 269) | 100 (37.17) | 82 (40.0) | 18 (28.1) | 0.08 | |
| Memory impairment (n = 268) | 74 (27.61) | 59 (28.9) | 15 (23.4) | 0.3 | |
| Neck stiffness (n = 269) | 49 (18.22) | 33 (15.6) | 16 (24.2) | 0.1 | |
| Vomiting (n = 269) | 54 (20.07) | 42 (20.5) | 12 (18.8) | 0.7 | |
| Convulsions (n = 269) | 23 (8.6) | 13 (6.3) | 10 (15.6) | 0.02 | 2.7 (1.01–7.1) |
| Vertigo (n = 269) | 32 (11.9) | 23 (11.2) | 9 (14.1) | 0.5 | |
| Brudzinski sign (n = 269) | 15 (5.58) | 8 (3.8) | 7 (10.6) | 0.05 | |
| Physical asthenia (n = 269) | 30 (11.1) | 23 (11.2) | 7 (10.9) | 0.9 | |
| Kernig sign (n = 269) | 14 (5.2) | 9 (4.2) | 5 (7.6) | 0.3 | |
| Visual disturbances (n = 269) | 8 (2.97) | 3 (1.5) | 5 (7.8) | 0.02 | 5.6 (1.04–37.5) |
| Functional Impairment (n = 269) | 17 (6.32) | 12 (5.9) | 5 (7.8) | 0.5 | |
| Clear CSF appearance (n = 265) | 249 (93.9) | 200 (94.3) | 62 (93.9) | 1 | |
| Raised CSF opening pressure (cm $H_2O$) (n = 92) | 65 (70.6) | 7 (10.6) | 17 (65.4) | <0.0001 | |
| Antiretroviral therapy (ART) (n = 278) | 204 (73.4) | 153 (72.2) | 51 (77.3) | 0.4 | |
| Poor outcome[3] (n = 217) | 78 (35.9) | 57 (35.4) | 21 (37.5) | 0.7 | |

[1]according to available data.

[2]column per cent calculated for each group.

[3]Death, status quo, discharge against medical advice, or transfer due to complications.

present convulsions (OR 2.7; IC 95%: 1.01–7.1, *p* = 0.02); and six times higher probability to present visual disturbances than no NMC patients (OR 5.6; IC 95%: 1.04–37.5, *p* = 0.02). The vast majority of NMC patients had clear CSF (93.9%) and a significantly raised CSF opening pressure (65.4%, *p*<0.0001). Furthermore, the poor therapeutic outcome was not significantly different between NMC and non-NMC patients (37.5 versus 35.4%, respectively).

## Biological analysis

**Routine diagnostic analysis of NMC.** Out of 66 NMC samples confirmed, 63 (95.5%, 95% CI: 89.4–100) had detectable cryptococcal antigen, only 29 (43.3%, 95% CI: 31.8–56.1) had yeasts presence after India ink staining, and the repeated culture was positive only in 43.3% of the cases (29/66, 95% CI: 31.8–56.1). All three CrAg negative samples were recovered from positive cultures.

**MALDI-TOF MS, ITS sequencing, and PCR serotyping characterization.** Of the 29 positive cultures, MALDI-TOF MS identified 23 as *C. neoformans* (79.3%), four as *Cutaneotrichosporon curvatus* (previously called *Cryptococcus curvatus*) (13.8%), and two (6.9%) could not be identified. While only 23 isolates were identified as serotype A using PCR serotyping (79.3%), six isolates were not serotypable (20.7%). All isolates were identified by ITS sequencing as follows: *C. neoformans* 79.3% (23/29), *Cutaneotrichosporon curvatus* 17.2% (5/29), and *Papiliotrema laurentii* (*Cryptococcus laurentii*) 3.5% (1/29). The results of the MALDI-TOF MS, ITS sequencing, and PCR serotyping characterization are summarized in Table 2. Additionally, MALDI-TOF MS detailed findings are presented in Table 3.

**MLST results.** Apart from the six strains identified as *Cutaneotrichosporon curvatus* (five) and *Papiliotrema laurentii* (one), the remaining 23 *Cryptococcus neoformans* isolates belong to the molecular type VNI. MLST analysis identified seven different STs: ST93 (15 isolates, 65.2%), ST5 (two isolates, 8.6%), ST53 (one isolate, 4.3%), ST31 (one isolate, 4.3%), ST4 (one isolate, 4.3%), ST69 (one isolate, 4.3%), and one novel ST that was not yet reported in the online fungal MLST database and was later assigned as ST659 (two isolates, 8.6%).

The application of the geoBURST algorithm to *Cryptococcus neoformans* (ST) isolates in the study identified two clonal complexes (CCs): CC1 (ST31 and ST93, as well as ST32 which was isolated 30 years earlier in the DRC) and CC2 (ST5 and ST53); and three singletons: ST659, ST4 and ST69 (Fig 1).

**Antifungal susceptibility test.** Overall, only the single strain of *Papiliotrema laurentii* was found to be resistant to amphotericin B (1/29, 3.4%). Concerning 5-flucytosine, two of 29 strains exhibited high MICs (6.9%), including the *Papiliotrma laurentii* strain and a strain of *Cryptococcus neoformans*. About 4/29 (13.8%) strains were categorized as resistant to fluconazole, including two *Cutaneotrichosporon curvatus* and two *Cryptococcus neoformans* strains. Considering only *Cryptococcus neoformans* strains, fluconazole resistance (MIC value > 8 mg/ L) was identified in 8.6% (2/23) of the cases.

**Table 2. MALDI-TOF MS, ITS sequencing, and multiplex PCR serotyping characterization.**

| Analysis | n = 29 (%) |
|---|---|
| MALDI-TOF MS | |
| *Cryptococcus neoformans* | 23 (79.3) |
| *Cutaneotrichosporon curvatus* | 4 (13.8) |
| Not identified | 2 (6.9) |
| ITS sequencing | |
| *Cryptococcus neoformans* | 23 (79.3) |
| *Cutaneotrichosporon curvatus* | 5 (17.2) |
| *Papiliotrema laurentii* | 1 (3.5) |
| Serotyping PCR | |
| Serotype A | 23 (79.3) |
| No identifiable | 6 (20.7) |

**Table 3. MALDI-TOF MS detailed findings.**

| Strain ID | Best match organism (score value) | Second-best match organism (score value) | Third-best match organism (score value) | NCBI identifier of the best match organism |
|---|---|---|---|---|
| BZ-3NGA | *Cryptococcus neoformans* P152 CBS (2.0) | *Cryptococcus neoformans* CBS 8710T CBS (1.81) | *Cryptococcus neoformans* CBS 996 CBS (1.81) | 178876 |
| BZ-6NGA | *Cryptococcus neoformans var. grubii* CBS 10512 CBS (1.59) | *Cryptococcus neoformans var. grubii* CBS 8710 CBS (1.46) | *Cryptococcus neoformans var. grubii* P152 CBS (1.40) | 178876 |
| BZ-9NGA | *Cryptococcus neoformans var. grubii* P152 CBS (1.85) | *Cryptococcus neoformans var. grubii* CBS 996 CBS (1.67) | *Cryptococcus neoformans* CBS 8710 CBS (1.58) | 178876 |
| BZ-13NGA | *Cryptococcus neoformans var. grubii* CBS 996 CBS (2.01) | *Cryptococcus neoformans var. grubii* P152 CBS (1.97) | *Cryptococcus neoformans* ICB 178 CBS (1.74) | 178876 |
| BZ-24NGA | *Cryptococcus neoformans var. grubii* CBS 996 CBS (1.74) | *Cryptococcus neoformans var. grubii* P152 CBS (1.60) | *Cryptococcus neoformans* CBS 10512 CBS (1.49) | 178876 |
| BZ-42NGA[1] | *Brevundimonas nosdae* DSM 14572T HAM (1.45) | *Strepmyces griseus* B261 UFL (1.16) | *Millerozyma farinozo* DSM 2226T DSM (1.16) | 33069 |
| BZ-44NGA | *Cryptococcus neoformans* P152 CBS (1.75) | *Cryptococcus neoformans* CBS 8710T CBS (1.53) | *Cryptococcus neoformans* ICB 165 CBS (1.45) | 178876 |
| BZ-46NGA | *Cryptococcus neoformans var. grubii* CBS 10512 CBS (1.53) | *Cryptococcus neoformans var. grubii* CBS 996 CBS (1.51) | *Cryptococcus neoformans var. grubii* ICB 165 CBS (1.50) | 178876 |
| BZ-73NGA | *Cryptococcus neoformans var. grubii* P152 CBS (1.74) | *Cryptococcus neoformans var. grubii* ICB 178 CBS (1.69) | *Cryptococcus neoformans var. grubii* CBS 996 CBS (1.55) | 178876 |
| BZ-94NGA[2] | *Pseudomonas viridiflava* DSM 11124T HAM (1.26) | *Cellulocimicrobium cellulans* B480 UFL (1.19) | *Pseudomonas pertucinogena* LMG 1874T HAM (1.19) | 33069 |
| BZ-97NGA | *Cryptococcus neoformans* P152 CBS (1.99) | *Cryptococcus neoformans* 29 PSB (1.86) | *Cryptococcus neoformans* ICB165 CBS (1.82) | 178876 |
| BZ-103NGA | *Cryptococcus neoformans* P152 CBS (1.89) | *Cryptococcus neoformans* ICB 165 CBS (1.63) | *Cryptococcus neoformans* CBS 996 CBS (1.55) | 178876 |
| BZ-105NGA | *Cryptococcus curvatus* CBS 570T CBS (1.51) | *Clostridium Haemolyticum* 1069 ATCC 9650T BOG (1.46) | *Rhizobium radiobacter* DSM 30147T HAM (1.28) | 57679 |
| BZ-110NGA | *Cryptococcus neoformans* P152 CBS (1.83) | *Cryptococcus neoformans* CBS 996 CBS (1.69) | *Cryptococcus neoformans* CBS 8710T CBS (1.62) | 178876 |
| BZ-124NGA | *Cryptococcus neoformans* P152 CBS (1.84) | *Cryptococcus neoformans* CBS 8710T CBS (1.81) | *Cryptococcus neoformans* CBS 10085 CBS (1.79) | 178876 |
| BZ-4RB | *Cryptococcus neoformans* P152 CBS (2.0) | *Cryptococcus neoformans* CBS 996 CBS (1.82) | *Cryptococcus neoformans* CBS 8710T CBS (1.75) | 178876 |
| BZ-12RB | *Cryptococcus neoformans* P152 CBS (1.80) | *Cryptococcus neoformans* CBS 8710T CBS (1.69) | *Cryptococcus neoformans* CBS 10512 CBS (1.62) | 178876 |
| BZ-22RB | *Cryptococcus neoformans* P152 CBS (1.78) | *Cryptococcus neoformans* CBS 10512 CBS (1.75) | *Cryptococcus neoformans* CBS 10085 CBS (1.63) | 178876 |
| BZ-27RB | *Cryptococcus neoformans var. grubii* CBS 10512 CBS (1.41) | *Cryptococcus neoformans var. grubii* ICB 178 CBS (1.35) | *Cryptococcus neoformans var. grubii* P152 CBS (1.32) | 178876 |
| BZ-28RB | *Cryptococcus neoformans var. grubii* P152 CBS (1.81) | *Cryptococcus neoformans var. grubii* CBS 10512 CBS (1.72) | *Cryptococcus neoformans var. grubii* CBS 996 CBS (1.60) | 178876 |
| BZ-55RB | *Cryptococcus neoformans* P152 CBS (2.02) | *Cryptococcus neoformans* CBS 8710T CBS (1.81) | *Cryptococcus neoformans* CBS 996 CBS (1.73) | 178876 |
| BZ-68RB | *Cryptococcus neoformans* CBS 996 CBS (1.69) | *Cryptococcus neoformans* P152 CBS (1.68) | *Cryptococcus neoformans* CBS 8710T CBS (1.57) | 178876 |
| BZ-78RB | *Cryptococcus neoformans* P152 CBS (1.85) | *Cryptococcus neoformans* CBS 8710T CBS (1.61) | *Cryptococcus neoformans* CBS 996 CBS (1.45) | 178876 |
| BZ-88RB | *Cryptococcus neoformans var. grubii* P152 CBS (1.60) | *Cryptococcus neoformans var. grubii* P10512 CBS (1.50) | *Cryptococcus neoformans var. grubii* ICB 165 (1.41) | 178876 |
| BZ-91RB | *Cryptococcus neoformans* CBS 8710T CBS (1.75) | *Cryptococcus neoformans* P152 CBS (1.55) | *Cryptococcus neoformans* 29 PSB (1.46) | 178876 |
| BZ-14LU | *Cryptococcus neoformans var. grubii* P152 CBS (1.54) | *Cryptococcus neoformans var. grubii* CBS 10085 CBS (1.54) | *Cryptococcus neoformans var. grubii* CBS 10512 CBS (1.48) | 178876 |

*(Continued)*

**Table 3.** (Continued)

| Strain ID | Best match organism (score value) | Second-best match organism (score value) | Third-best match organism (score value) | NCBI identifier of the best match organism |
|---|---|---|---|---|
| BZ-23LU | *Cryptococcus curvatus* CBS 570T CBS (1.70) | *Cellulocimicrobium cellulans* B480 UFL (1.36) | *Cryptococcus neoformans* ICB175 SDA CBS (1.28) | 57679 |
| BZ-29LU | *Cryptococcus curvatus* CBS 570T CBS (1.52) | *Pseudomonas rhizosphaerae* LMG 21640T HAM (1.39) | *Agromyces cerinus ssp cerinus* HKI 11525_DSM 8595T HKJ (1.28) | 57679 |
| BZ-33LU | *Cryptococcus curvatus* CBS 570T CBS (1.91) | *Cryptococcus curvatus* CBS 5163 CBS (1.46) | *Pseudomonas graminis* DSM 11363T HAM (1.36) | 57679 |

[1]Strain later identified as *Cryptococcus laurentii*.

[2]Strain later identified as *Cryptococcus curvatus*.

### NMC severity factors and MLST ST of *Cryptococcus neoformans* isolates

Among NMC severity factors previously described and those considered in the present study, only the poor therapeutic outcome was associated with infections due to the less common MLST STs isolates (ST5, ST659, ST53, ST31, ST4, and ST69) versus the main ST (ST93) (87.5% vs. 40%, respectively; $p = 0.02$). Table 4 summarizes the NMC severity factors compared to the MLST ST of *C. neoformans* isolates.

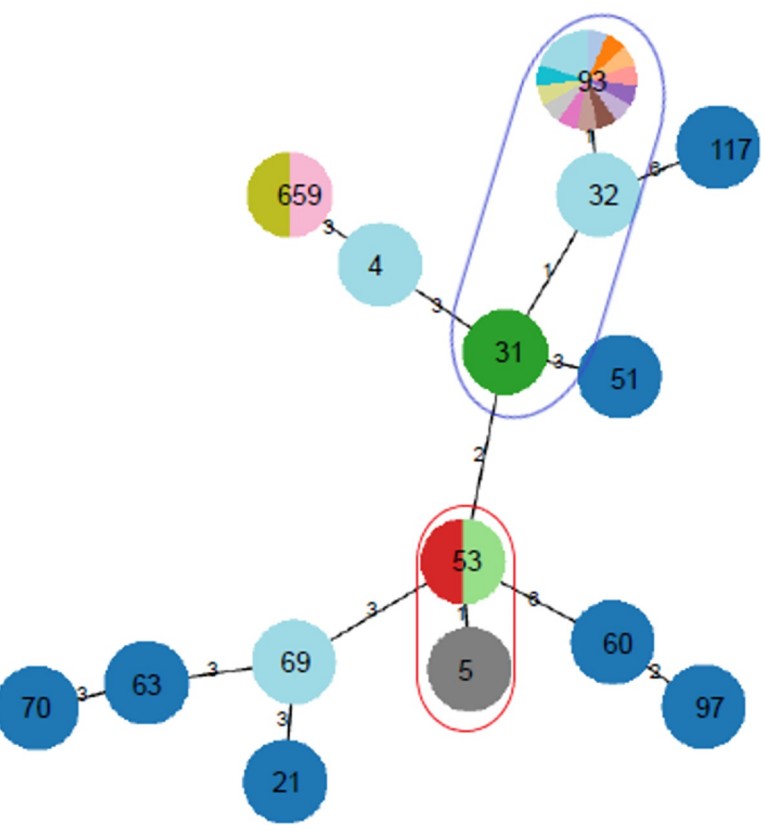

**Fig 1. Minimum Spanning Tree showing the distribution of the 23 *Cryptococcus neoformans* (STs, VNI) isolates from advanced HIV disease patients using geoBURST analysis.** Clonal colours indicate the source of the isolates, distinguishing the study isolates, former DRC isolate, and the reference isolates. Blue circle shows CC1 and the red circle shows CC2.

**Table 4. NMC severity factors compared to the MLST STs of *C. neoformans* isolates.**

| Variable | *Cryptococcus neoformans* ST identified | | p |
|---|---|---|---|
| | Main study ST[1] n[3] (%)[4] | Less common STs[2] n[3] (%)[4] | |
| Glycorhachia (mg/dl) (n = 10) | | | 1 |
| Low (≤ 50) | 7 (87.5) | 2 (100) | |
| High (≥ 60) | 1 (12.5) | 0 | |
| Opening pressure (cm $H_2O$) (n = 8) | | | 0.4 |
| Moderately high (<30) | 2 (40) | 0 | |
| Very high (≥30) | 3 (60) | 3 (100) | |
| Therapeutic outcome (n = 23) | | | 0.02 |
| Good[5] | 9 (60) | 1 (12.5) | |
| Poor[6] | 6 (40) | 7 (87.5) | |

[1]ST93.

[2]STs5, 659, 53, 31, 4, 69.

[3]With available data.

[4]Percentage of columns calculated for each group.

[5]Recovery and discharge from hospital.

[6]Death, status quo, discharge against medical advice, or transfer due to complications.

## Discussion

We described the clinical epidemiology of neuromeningeal cryptococcosis (NMC) in advanced HIV disease patients (AHIVP) admitted to the tertiary facilities, as well as the results of antifungal susceptibility testing and molecular characterization of *Cryptococcus* spp. isolates. In addition, we investigated the association between NMC severity factors and *Cryptococcus neoformans* MLST sequence types.

Neuromeningeal cryptococcosis (NMC) prevalence in AHDP was estimated at 23.7% (95% CI: 18.7–28.8). This prevalence is much higher than that reported in France [22], in other African countries [23–26], and previously in the DRC [4]. However; it is approximately similar to that described in Cameroon [27] and slightly lower than that reported in Kenya [28]. This could be explained by (a) differences in HIV infection prevalence in various countries and regions, (b) the HIV management and opportunistic infections prevention policy applied in each country, (c) characteristics of the patient population included in each study, (d) the sensitivity and specificity of diagnostic tests used, and (e) the species/molecular types/sequences type circulating in each region. Female patients aged 42.19 ± 12.13 years old, married/cohabitating and with secondary education level were (non-significantly) most affected. This trend is consistent with the demographic profile of PLHIV in the DRC. McClelland E. *et al.* found that virulent *C. neoformans* phenotypes from females had longer doubling times and released more capsular glucuronoxylomannan (GXM) in the 17-β estradiol presence. Additionally, macrophages from women phagocytized more *C. neoformans* than those from men. Males hence had a higher fungal load than females and their macrophages were more likely to be destroyed by *C. neoformans* [29]. This protective trend in women was not significantly noted in the current study. Headache, convulsions and visual disturbances were significantly associated with NMC. These data are largely consistent with the literature [30]. Visual disturbances are known to be associated with NMC in 18% of cases following raised intracranial pressure, which is slightly more than the proportion reported in this study (7.7%) [31]. Among the clinical symptoms and signs (headache, depressed sensorium, papilledema, and raised CSF opening pressure)

and radiological findings (flattening of the posterior sclera, increased CSF in the subarachnoid space around the optic nerve, optic nerve tortuosity and empty parietal saddle), only headaches and raised CSF opening pressure were found in the present study [31]. NMC was associated with higher raised CSF opening pressure, CD4 count < 100 cells/mm$^3$ and HIV-infection stage IV. One of the most critical outcome determinants in PLHIV with NMC is the raised opening CSF pressure which is generally correlated with high CSF fungal load, co-morbidities and increased risk of death [32]. In agreement with the data described by Bicanic *et al*, 63% of patients in the present study had a very high opening CSF pressure [33].

Cryptococcal antigen (CrAg) detection was the main diagnostic tool for NMC in the present study (95% positivity rate out of 66 confirmed samples). Compared to other studies, culture positivity rate, and *Cryptococcus* India ink staining identification were very low [34]. For Kabanda T. *et al*, CrAg detection has shown more sensitivity in clinical situations (100%) than culture (95.7%) and India ink staining (93.6%) [35]. The low positivity rate of direct microscopy and culture found in this study could be caused by the suboptimal storage conditions of samples before analyses and/or probable low CSF fungal load in certain samples [36].

ITS sequencing yielded an identification for all included strains (29/29). Described as the main responsible species for cryptococcosis diseases in PLHIV [37], *C. neoformans* was identified in 73.3% of all isolates. The remaining cases were identified as *Cutaneotrichosporon curvatus* (17.2%) and *Papiliotrema laurentii* (3.5%). Initially considered saprophytic and non-pathogenic to humans, these last two non-*Cryptococcus* species previously described as non-*neoformans/gattii Cryptococcus* species, are becoming increasingly common in clinical infections [7,10,38]. Although the clinical presentation of *C. neoformans* NMC is more severe than in non-*Cryptococcus* NMC, some *Cutaneotrichosporon curvatus* and *Papiliotrema. laurentii* strains exhibited resistance to common antifungal agents, such as 5-flucytosine and amphotericin B. In addition, these species are more difficult to identify by routine laboratory methods than *C. neoformans* [7]. In this study, out of six non-*Cryptococcus* species, only four were correctly identified by MALDI-TOF MS compared to the identification obtained using ITS sequencing. The lack of certain species reference spectra in the database provided by the mass spectrometry manufacturer (reference library MBT Filamentous Fungi Library 3.0, Bruker Daltonics) for identification, and genome differences between *Cryptococcus* and non-*Cryptococcus* species which would result in huge differences in proteomic profiles between these species, could explain these results. Inconclusive MALDI TOF MS identifications were therefore associated with poor MALDI TOF MS scores. Serotype A isolates were previously reported to be the most commonly isolated serotypes in environmental and clinical settings [39], both worldwide and in DRC, a trend also confirmed in the current study.

The MLST analysis of *C. neoformans* isolates revealed a large heterogeneity of STs within the single molecular type found in the present study (VNI), involving seven distinct STs. In line with our results, *Cryptococcus neoformans* ST93 is the most isolated in various countries (China, India, Indonesia, South Africa, Thailand, Brazil, Uganda, and Colombia), both in clinical and environmental settings [40–45]. This ST has been associated with high mortality in Uganda [43]. However, the STs distribution amongst *Cryptococcus neoformans* VNI population in Ivory Coast is dominated by ST5, while ST93 is only found in 1.3% of cases [46]. In Nigeria, ST32 has been repeatedly reported [47]. Therefore, each region should identify the genetic profile of the strains circulating and determine epidemiological, clinical and therapeutic implications, in order to adapt regional management guidelines.

In this study, ST93 was associated with better treatment outcomes than the less common STs. This opens up the debate on the relative virulence of each ST compared to the others. Although most of the STs had already been isolated in the DRC neighbouring countries and recorded in the International Fungal MLST Database, namely ST4 in Uganda and Tanzania,

ST5, ST31, ST69, and ST93 in Uganda only. One ST (ST53) had previously been isolated only in Thailand and in no other country worldwide (https://mlst.mycologylab.org/) [45]. In addition, one isolate had an ST identified for the first time in the present study, subsequently assigned as ST659. STs analysis based on the geoBURST algorithm revealed a strong link between ST32 (a Congolese strain isolate more than 30 years earlier) and the CC1 identified in this study. This could suggest a descending evolution between these STs.

Overall, one isolate of *Papiliotrema. laurentii* was resistant to amphotericin B and 5-flucytosine, and four strains had high MICs values for fluconazole, including two *Cryptococcus neoformans* and two *Cutaneotrichosporon curvatus*. These results may partly explain the high mortality rates previously described in Congolese studies [4,48]. With triple antifungal therapy routinely administered to all NMC patients, the antifungal resistances described here did not significantly affect patients' outcomes. Nevertheless, large-scale evaluations should be conducted and treatment regimens updates should be made if necessary.

As described by Trilles *et al.* regarding the low antifungal susceptibility of VGI isolates compared to the other molecular types tested [49], the less common STs isolates identified in the present study were associated with poor therapeutic outcomes.

## Conclusions

Using a combination of diagnostic tests, we report a higher epidemiological burden of NMC among symptomatic AHDP in DRC than previously described. The following fungal species were identified in CSF samples: *Cryptococcus neoformans*, *Cutaneotrichosporon. curvatus* and *Papiliotrema. laurentii*. MLST analysis demonstrated a variety of sequence types within the only molecular type found (VNI), including one major ST (ST93) and six less common STs (ST5, ST659, ST53, ST31, ST4, and ST69). We report a fluconazole resistance rate of 13.8% (4/29), in addition to strains with sporadic resistance to amphotericin B and 5-flucytosine; this requires careful monitoring. Additionally, we find that non-ST93 sequence types were associated with poor therapeutic outcomes. Larger studies are required to consolidate the findings from this study.

## Supporting information

**S1 File. CryptoDepositSheetNov2020_Bive (1).**
(XLSX)

**S2 File. Cutaneotrichospora curvatus-Papiliotrema laurentii ITS-2.**
(DOCX)

**S3 File. ITS 2 Cryptococcus spp. sequences_PLOS ONE.**
(FASTA)

## Acknowledgments

The authors sincerely thank Dr Klaas Dewaela for the editing and proofreading of the English language.

## Author Contributions

**Conceptualization:** Bive Zono Bive.

**Data curation:** Bive Zono Bive, Hippolyte Situakibanza Nani-Tuma, Pius Kabututu Zakayi.

**Formal analysis:** Bive Zono Bive, Rosalie Sacheli, Alex Ka, Raphael Boreux, Wieland Meyer.

**Funding acquisition:** Georges Mvumbi Lelo, Marie-Pierre Hayette.

**Investigation:** Bive Zono Bive.

**Methodology:** Bive Zono Bive.

**Resources:** Gaultier Muendele, Nicole Landu, Pierre-Robert M'Buze.

**Software:** Alex Ka.

**Supervision:** Hippolyte Situakibanza Nani-Tuma, Georges Mvumbi Lelo, Marie-Pierre Hayette.

**Validation:** Hippolyte Situakibanza Nani-Tuma, Pius Kabututu Zakayi, Marcel Mbula Mambimbi, Raphael Boreux, Celestin Nzanzu Mudogo, Michel Moutschen, Wieland Meyer, Georges Mvumbi Lelo, Marie-Pierre Hayette.

**Writing – original draft:** Bive Zono Bive.

**Writing – review & editing:** Rosalie Sacheli, Hippolyte Situakibanza Nani-Tuma, Pius Kabututu Zakayi, Alex Ka, Marcel Mbula Mambimbi, Gaultier Muendele, Raphael Boreux, Nicole Landu, Celestin Nzanzu Mudogo, Pierre-Robert M'Buze, Michel Moutschen, Wieland Meyer, Georges Mvumbi Lelo, Marie-Pierre Hayette.

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
