## [Decision Letter · Decision Letter 0]

18 Feb 2022

PONE-D-21-40474Clinical epidemiology and high genetic diversity amongst Cryptococcus spp. isolates infecting people living with HIV in Kinshasa, Democratic Republic of CongoPLOS ONE

Dear Dr. ZONO,

Thank you for submitting your manuscript to PLOS ONE. After careful consideration, we feel that it has merit but does not fully meet PLOS ONE’s publication criteria as it currently stands. Therefore, we invite you to submit a revised version of the manuscript that addresses the points raised during the review process.

Although well conducted, the study requires further experiments; in particular the Authors should evaluate the susceptibility of strains to antifungal agents. Furthermore, the manuscript requires a strong revisions concerning English Language and an update of the bibliography.

We look forward to receiving your revised manuscript.

Kind regards,

Adriana Calderaro

Academic Editor

PLOS ONE

Journal Requirements:

4. We noted in your submission details that a portion of your manuscript may have been presented or published elsewhere. 

( It is noteworthy that comparative data between PLHIV with Cryptococcus neoformans versus Cryptococcus curvatus/ C. laurentii meningitis are presented in another published paper (https://doi.org/10.1186/s12879-021-06849-3). Although carried out in the same study population by the same research team, the data are presented separately without any interference)

Reviewers' comments:

Reviewer's Responses to Questions

**Comments to the Author**

1. Is the manuscript technically sound, and do the data support the conclusions?

Reviewer #1: Yes

Reviewer #2: Yes

2. Has the statistical analysis been performed appropriately and rigorously? 

Reviewer #1: I Don't Know

Reviewer #2: No

3. Have the authors made all data underlying the findings in their manuscript fully available?

Reviewer #1: Yes

Reviewer #2: Yes

4. Is the manuscript presented in an intelligible fashion and written in standard English?

Reviewer #1: Yes

Reviewer #2: Yes

5. Review Comments to the Author

Reviewer #1: This is a well written manuscript that addresses a pathogen that continues to cause high morbidity and mortality among PLHIV with advanced disease. What i see missing are implication of this work programmatically. Nonetheless, this is a well written review

Reviewer #2: The manuscript by Zono Bive et al. deals with the genetic diversity of Cryptococcus strains isolated in the Democratic Republic of Congo. They use ITS sequencing, MALDI-TOF and MLST. The article is well conducted despite a limited number of strains. The structure of the article is logical and well built. However, the English language could be proofread. An evaluation of the susceptibility of the tested strains is also missing.

Concerning the remarks, they are essentially general:

- The new nomenclature of Cryptococcus laurentii and Cryptococcus curvatus (i.e. Cutaneotrichospora and Papiliotrema) should be used (Liu 2015, Sujita 2017)

- The epidemiological data used (Park 2009) are obsolete. Those published by Rajasingham in 2017 (ref. 1) should be cited

- p 6. The 194 correct 55.8

- Tables should be reformatted according to journal standards

- Regarding Table 2, the MALDI-TOF scores should be specified to know the reliability of the identifications. This could explain the wrong identification of Papiliotrema.Percentages are not really useful in this table.

- Figure 1 is not appropriate. It is necessary to generate a Minimum Spanning Tree and to place the STs in a global environment allowing to determine CCs.

- There is a serious lack of data on the susceptibility of strains to major antifungal drugs. This would strengthen this study and strongly support the Discussion.

- In the Discussion section (p10. L 266-268): the references are too old (ref. 18) and the choice of countries is not relevant. There are very good studies on cryptococcosis in Africa and they deserve to be added to your Discussion.For example, there are studies in Cameroon, Côte d'Ivoire or Kenya (Loyse 2019, Kassi 2019, Gitonga 2019).

- P10. L 268-272: ST can also explain your hypothesis.

- p11. L 300-302: C. laurentii has long been known to cause infections in humans (Kamalam 1976, Lynch 1981).

- P 11. L 313-314: Here again, it is imperative to compare the data with those published in other African countries (e.g. ST 5 and 93 are present in Ivory Coast).

-P 11. L317-322: please cite references

6. PLOS authors have the option to publish the peer review history of their article (what does this mean?). If published, this will include your full peer review and any attached files.

Reviewer #1: No

Reviewer #2: No

---

## [Author Response · Author response to Decision Letter 0]

19 Mar 2022

Dear Editor, 

First of all, I would like to thank you and the reviewers of this manuscript through you, who have clearly and significantly contributed to its improvement.

As mentioned in the manuscript, this work is from the cross-sectional study conducted by our team in tertiary hospitals in Kinshasa (DRC). The initial results led to the elaboration of a comparative study of the baseline clinical and biological profiles of patients with Cryptococcus neoformans versus non-neoforman/non-gattii (non-Cryptococcus) meningitis, exclusively the 29 patients who had a positive culture. This article has been peer-reviewed and published online: https://doi.org/10.1186/s12879-021-06849-3.

In the current manuscript, we discuss the general results of this epidemiological investigation (cross-sectional study), based on the clinical profile of the included subjects, routine biological data, molecular identification (PCR serotyping, MALDI TOF MS, ITS sequencing, MLST characterization) and antifungal susceptibility of Cryptococcus spp. isolates. All patients’ data are included here. These are therefore not the same data profiles. 

In the following lines, you will find our detailed answers.

Point-by-point response

I. Academic editor’ comments

- Abstract: Could do with a line on implications for programs.

R/ Lines on the implications for programs have been added in the introductory part of the manuscript. Lines 129-133

- Introduction: Given the size of DRC, could do with some further data that show the variability by geography 

R/ The additional data on geographical variability in the DRC has been added in the text. Lines 93-97

- Results: Would have been important to include some cascade data that starts with all who were hospitalized, how many had suspected cryptococcal disease and how many had a lumbar puncture. 

R/ Following the suggestion, we specified here that all included patients (278) were hospitalized and had a lumbar puncture for investigation. Lines 246-250

- A couple of grammatical areas to look at. Line 189 and 190: “men patients” and “women patients” seems odd. Would rather use, “men” and “women”. 

R/ Changes made in the manuscript. Lines 248-250

- Line 195- “stage before NMC diagnosis, NMC tends to develop during HIV-infection stage IV (95.5%, p = 0.0008)”. Would use “tended” instead of “tends”. 

R/ Changes made in the text. Line 255

- Line 196: Use four times “higher” probability. 

R/ Changes made in the manuscript. Lines 256-259

- In general some sentences are written in present tense while others are presented in past tense.

R/ Uniformity has been revised in the text.

Discussion: 

- Line 263: “We described the clinical epidemiology of PLHIV” could be re-written to specify that the authors described the clinical epidemiology of PLHIV with advanced HIV admitted to a tertiary facility and not all PLHIV.

R/ Changes made in the text. Lines 329-331

II. Reviewer 1’ comments 

- What I see missing are implication of this work programmatically.

R/ The programmatic implications of this work have been incorporated into the manuscript. Lines 129-133

III. Reviewer 2’ comments

3.1. Main comment

- The English language could be proofread. 

R/ Based on the suggestion, the manuscript was reviewed and edited by a native English-speaking.

- An evaluation of the susceptibility of the tested strains is also missing.

R/ The susceptibility of the strains to antifungal agents was tested and included in the manuscript.

3.2. General remarks

- The new nomenclature of Cryptococcus laurentii and Cryptococcus curvatus (i.e. Cutaneotrichospora and Papiliotrema) should be used (Liu 2015, Sujita 2017).

R/ The new nomenclature has been updated in the manuscript and the corresponding references were also updated.

- The epidemiological data used (Park 2009) are obsolete. Those published by Rajasingham in 2017 (ref. 1) should be cited.

R/ Changes made in the manuscript. Lines 88-92

- p 6. The 194 correct 55.8

R/ Changes made in the text. Line 254

- Tables should be reformatted according to journal standards

R/ improvements have been made in the concerned files

- Regarding Table 2, the MALDI-TOF scores should be specified to know the reliability of the identifications. This could explain the wrong identification of Papiliotrema.

R/ A supplementary table containing details of the MALDI-TOF MS findings including scores has been added.

- Figure 1 is not appropriate. It is necessary to generate a Minimum Spanning Tree and to place the STs in a global environment allowing to determine CCs

R/ Minimum Spanning Trees using the geoBURST algorithm were generated and the corresponding figure has been added.

- There is a serious lack of data on the susceptibility of strains to major antifungal drugs. This would strengthen this study and strongly support the Discussion.

R/ The susceptibility of strains to major antifungal drugs was tested and discussed in the manuscript.

- In the Discussion section (p10. L 266-268): the references are too old (ref. 18) and the choice of countries is not relevant. There are very good studies on cryptococcosis in Africa and they deserve to be added to your Discussion. For example, there are studies in Cameroon, Côte d'Ivoire or Kenya (Loyse 2019, Kassi 2019, Gitonga 2019).

R/ Update done in the manuscript. For this discussion part, the first intention was to compare the hospital prevalence of NMC in HIV populations in different regions. Articles with this baseline data have been completed, including the great suggested articles that have been extensively used in the rest of the discussion. Lines 334-337

- P10. L 268-272: ST can also explain your hypothesis.

R/ Suggestion taken into account in the manuscript. Lines 338-342

- P11. L 300-302: C. laurentii has long been known to cause infections in humans (Kamalam 1976, Lynch 1981).

R/ Correction taken into account in the manuscript. Lines 374-378

- P 11. L 313-314: Here again, it is imperative to compare the data with those published in other African countries (e.g. ST 5 and 93 are present in Ivory Coast).

R/ References from studies conducted in African regions on MLST characterisation of Cryptococcus neoformans/Cryptococcus gattii strains were sought and considered. Lines 393-407

- P 11. L317-322: please cite references

R/ It should be noted here that all STs mentioned in this section were obtained from the online database: international fungal MLST database: https://mlst.mycologylab.org/. Details have been provided in the manuscript and corresponding references have been cited.

---

## [Decision Letter · Decision Letter 1]

18 Apr 2022

Clinical epidemiology and high genetic diversity amongst Cryptococcus spp. isolates infecting people living with HIV in Kinshasa, Democratic Republic of Congo

PONE-D-21-40474R1

Dear Dr. ZONO,

We’re pleased to inform you that your manuscript has been judged scientifically suitable for publication and will be formally accepted for publication once it meets all outstanding technical requirements.

Kind regards,

Adriana Calderaro

Academic Editor

PLOS ONE

Additional Editor Comments (optional):

Reviewers' comments:

Reviewer's Responses to Questions

**Comments to the Author**

1. If the authors have adequately addressed your comments raised in a previous round of review and you feel that this manuscript is now acceptable for publication, you may indicate that here to bypass the “Comments to the Author” section, enter your conflict of interest statement in the “Confidential to Editor” section, and submit your "Accept" recommendation.

Reviewer #1: All comments have been addressed

2. Is the manuscript technically sound, and do the data support the conclusions?

Reviewer #1: Yes

3. Has the statistical analysis been performed appropriately and rigorously? 

Reviewer #1: N/A

4. Have the authors made all data underlying the findings in their manuscript fully available?

Reviewer #1: Yes

5. Is the manuscript presented in an intelligible fashion and written in standard English?

Reviewer #1: Yes

6. Review Comments to the Author

Reviewer #1: Excellent responses to the review comments. The paper reads well with a good description of the context, the impact of cryptococcosis and implications for clinical care clearly laid out.

7. PLOS authors have the option to publish the peer review history of their article (what does this mean?). If published, this will include your full peer review and any attached files.

Reviewer #1: No

---

## [Editor Report · Acceptance letter]

12 May 2022

PONE-D-21-40474R1 

Clinical epidemiology and high genetic diversity amongst *Cryptococcus* spp. isolates infecting people living with HIV in Kinshasa, Democratic Republic of Congo 

Dear Dr. Bive:

I'm pleased to inform you that your manuscript has been deemed suitable for publication in PLOS ONE. Congratulations! Your manuscript is now with our production department. 

Kind regards, 

on behalf of

MD, PhD, Associate Professor Adriana Calderaro 

Academic Editor

PLOS ONE